# Stigma processes, psychological distress, and attitudes toward seeking treatment among pedohebephilic people

**Sara Jahnke**[1,2]*, **Ian V. McPhail**[3], **Jan Antfolk**[2]

**1** Department of Health Promotion and Development, University of Bergen, Bergen, Norway, **2** Faculty of Arts, Psychology, and Theology, Åbo Akademi University, Turku, Finland, **3** Bloomberg School of Public Health, Johns Hopkins University, Baltimore, Maryland, United States of America

* sara.jahnke@uib.no

**Data Availability Statement:** The raw data and syntax are available for researchers upon request via the database surveybanken, provided by the Norwegian Agency for Shared Services in

## Abstract

While stigma theories predict that stigma relates to negative attitudes toward seeking help, previous studies found mixed results among pedohebephilic individuals. We tested whether different stigma processes (i.e., general anticipated stigma, anticipation of negative therapist behavior upon disclosure, and internalized stigma), psychological distress, previous treatment experiences, and knowledge about psychotherapy can clarify attitudes toward seeking professional psychological help for this population. We conducted a pre-registered, online survey of English-speaking pedohebephilic individuals ($N = 283$, 88% male). Expected links between variables of interest and attitudes toward seeking treatment were assessed via structural equation modeling. After modification, the final model showed acceptable fit to the data, $\chi^2 = 2170.61$, $df = 1462$, $p < .001$, CFI = .905, RMSEA = .04, [.04,.05]. Internalized stigma predicted more positive attitudes toward seeking treatment. Lower anticipation of negative therapist behavior upon disclosure and higher knowledge about psychotherapy were also significant predictors of positive attitudes toward seeking treatment. General anticipated stigma did not predict attitudes toward seeking treatment. Our results suggest a complex association between different stigma processes and treatment-seeking, which differs from associations found for mental illness stigma. Specificity in our understanding of the components of stigma and how they interact with attitudes toward seeking treatment is required to tailor clinical work as well as messages around treatment services.

## Introduction

Up to one percent of the male population are pedophilic [1, 2], meaning that they have a sexual attraction to prepubescent children [3]. Hebephilia, which connotes a sexual attraction to early pubertal children, is likely to be more common, with up to four percent of men reporting such interests [1, 2]. Women are considerably less likely to report pedophilia or hebephilic attraction than men [1]. Pedohebephilic individuals may be in need of treatment for various reasons,

Education and Research (Sikt). This is in line with information provided to participants during the informed consent procedures, as publication of the dataset is likely to deter pedohebephilic people from participating in research studies due to fears of being identified. Citation: Jahnke, Sara. (2024). Better therapy for people who are sexually attracted to children: A survey on labels, treatment motivation, and treatment experiences [Data set]. Sikt - Norwegian Agency for Shared Services in Education and Research. https://doi.org/10.18712/NSD-NSD3178-V2.

**Funding:** This study was funded by a personal grant (application number 333761) to the first author from the Academy of Finland. The funders had no role in study design, data collection and analysis, decision to publish, or preparation of the manuscript.

**Competing interests:** The authors have declared that no competing interests exist.

such as suicidal ideation [4]; depression, anxiety, or substance disorders [5]; coping with stigma [6]; or support to manage risk of committing sexual offenses [7]. Pedophilia and hebephilia are not considered pathological in and of themselves. Nevertheless, individuals with pedophilia who experience distress or interpersonal difficulties relating to their sexual interests may be eligible for a diagnosis of *pedophilic disorder* [3].

In line with previous literature reviews demonstrating a link between stigma and negative attitudes toward seeking treatment [8, 9], stigma related stressors may work to deter pedohebephilic individuals from seeking treatment [10, 11]. Yet, quantitative research has either failed to detect a link between stigma and attitudes toward seeking treatment [12, 13] or found that stigma is linked to a *more positive* attitudes toward seeking treatment in this population [14]. We assume that the links between different stigma processes and attitudes toward seeking treatment might go in opposite directions: (1) anticipating stigma from the therapist or having internalized stigma may push people away from seeking therapy, while (2) anticipating or internalizing stigma may increase the need for treatment because of the distressful nature of these experiences. Given the importance of reducing barriers to accessing mental health services, the present research aims to extend our understanding of the complex ways in which stigma relates to attitudes toward seeking treatment among pedohebephilic individuals.

## Attitudes toward seeking treatment and its link to different stigma processes

Various barriers exist for people seeking treatment. More negative attitudes toward seeking treatment are linked to less knowledge about psychotherapy, lower rates of psychological distress, and a lack of prior treatment experiences [15, 16]. Stigma represents another factor that may deter potential clients. Stigma refers to a characteristic that is subject to devaluation, the reactions of others towards those who possess the devalued characteristic, and the reactions of people who possess the stigmatized characteristic to themselves and to others [17]. Modern stigma theories treat the experience of stigma as a multifaceted construct [17] that includes experienced stigma, internalized stigma, and anticipated stigma [18]. Experienced stigma includes experiencing prejudice from others, such as stereotyping or discrimination. Anticipated stigma refers to expecting members of the dominant society to react negatively upon detecting the devalued characteristic, such as with social exclusion or rejection. Anticipating such responses can also contribute to efforts to conceal one's stigmatized identity in order to avoid negative reactions [19]. Internalized stigma is the process by which individuals who possess a devalued characteristic come to accept negative public beliefs and feelings about their group to be true of themselves [17, 18, 20]. Within the minority stress model [19], experienced, anticipated, and internalized stigma are understood as unique causes of distress and mental health problems for minority groups.

Pedohebephilic individuals are a highly stigmatized group who are acutely aware of the stigmatizing attitudes society has toward their sexual attractions [10, 12, 21]. Recent research suggests that pedohebephilic individuals are more highly stigmatized compared with individuals with other paraphilic interests [22] and individuals who abuse substances or behave in antisocial ways [23]. Yet certain stigma processes may operate in different ways for this population. For instance, pedohebephilic individuals tend to report low levels of experiencing discrimination and prejudice in their daily lives [24]. This may be due to concealment efforts and the concealability of pedohebephilic interests. There is, however, evidence that pedohebephilic individuals anticipate negative reactions by others in their lives [25], including therapists if they were to seek treatment [11, 26]. Internalizing stigma in this population tends to operate in

hypothesized ways and is associated with a range of negative psychosocial and mental health problems [4, 12, 25].

## Anticipated stigma and treatment seeking

Studies on the link between anticipating rejection and seeking treatment for a mental health problem have produced mixed results [8]. Prior research did not detect a significant link between general anticipation of negative reactions by unspecified others and attitudes toward seeking treatment in pedohebephilic individuals [12]. Yet, this stigma process was not assessed with reference to a potential therapist who would be the recipient of disclosure rather than a vague or general group. It is likely that a link between anticipating negative responses and attitudes toward seeking treatment will be identified if both concepts are assessed with a similar degree of specificity [27], such as when focusing on anticipated negative reactions from a potential therapist. One prior study confirmed that anticipation of *therapists* holding negative attitudes towards lesbian, gay, and bisexual (LGB) people predicted less positive attitudes toward seeking treatment among this group [28]. Similar processes may be at work among pedohebephilic individuals, as anticipation of being rejected or being unnecessarily reported to the authorities by clinicians are frequently reported barriers to seeking treatment [11, 26]. It is possible that a more multifaceted assessment of these constructs will identify links between these anticipations of stigma and attitudes toward seeking treatment.

## Internalized stigma and treatment seeking

Internalized stigma has been theorized to reduce positive attitudes toward seeking treatment by undermining people's belief in their ability to seek and to benefit from mental health services [20]. A systematic review found a small, negative association between internalized stigma and measures of attitudes, willingness, and behaviors related to seeking general mental health treatment [8]. Yet, in one study the opposite effect appears to hold for LGB individuals, with participants who report a more negative view of their LGB identity reporting *more positive* attitudes toward seeking treatment [28]. This leads to the speculation that for stigmas other than mental illness stigma, internalized stigma could improve attitudes toward seeking treatment as an opportunity to discuss such negative feelings in a supportive environment.

As many pedohebephilic individuals report wanting treatment to learn how to live with stigma [6], the way in which internalized stigma relates to attitudes toward seeking treatment may be similar to LGB stigma. Likewise, internalized stigma and concealment among pedohebephilic individuals were associated with greater motivation to make improvements in their lives, including seeking treatment services [14]. In contrast, Lievesley et al. [13] found no link between (general) feelings of shame (as a proxy for internalized stigma) and past help-seeking ($r = -.04$) or need for further support ($r = -.01$), but they did not directly assess internalized stigma. In another survey of pedohebephilic individuals, approximately half of the sample indicated that believing news media sources that individuals with these attractions could not be helped was a barrier to seeking treatment, suggesting internalizing certain features of stigmatizing attitudes may reduce treatment seeking [11]. While there is some preliminary evidence for the assumed link between internalized stigma and attitudes toward seeking treatment for this population, the evidence is not entirely clearcut. Further, this pathway may include a mediating variable. Past research with general mental health populations and pedohebephilic individuals finds an association between internalized stigma and psychological distress [25, 29]. Given the association between distress and treatment seeking [30], it is conceivable that psychological distress and reduced wellbeing mediate the pathway between internalized stigma

and general anticipated stigma on the one side, and more positive attitudes toward seeking treatment on the other side.

## The present study

We hypothesized that different stigma processes show differential relations to attitudes toward seeking treatment. Other relevant predictors (e.g., knowledge about psychotherapy) and mediators (e.g., psychological distress) were also examined. Note that while our hypotheses use the term "predict" in the context of statistical modelling, causality cannot be inferred because of the cross-sectional and observational nature of our data.

The following binary hypotheses were pre-registered: We assumed zero-order associations between positive attitudes toward seeking treatment and 1) knowledge about psychotherapy, 2) general anticipated stigma, 3) internalized stigma, 4) anticipated negative therapist behavior upon disclosure, 5) psychological distress, and 6) psychological wellbeing. For anticipated negative therapist behavior, we specified that this link was expected to be negative. Additionally, we assumed that psychological wellbeing and psychological distress are linked to 7) general anticipated stigma (positive correlation for psychological wellbeing and negative correlation for psychological distress), and 8) internalized stigma (positive correlation for psychological wellbeing and negative correlation for psychological distress). Furthermore, we pre-registered zero-order correlations between anticipated therapist maltreatment upon disclosure and 9) Knowledge about psychotherapy (negative association).

It was also pre-registered that these associations (1–9) will be tested using structural equation modeling (SEM), alongside two potential indirect effects: 1) if the putative effect of knowledge about psychotherapy on attitudes toward seeking treatment is mediated via anticipated negative therapist behavior upon disclosure, and 2) if the putative effects of general anticipated stigma and internalized stigma on attitudes toward seeking treatment are mediated by psychological distress and reduced wellbeing.

## Materials and methods

### Participants

The study was approved by the Board for Research Ethics at Åbo Akademi University. Recruitment took place between 7th of January and 6th of May 2021. Hence, data collection was stopped exactly one month short of the 5 months that were pre-registered as the stopping rule due to an honest mistake on the part of the first author, not a decision based on the data or results. The link to the survey was posted on English (B4U-ACT, BoyChat, Virtuous Pedophiles, VisionsofAlice) and German-language forums (jungsforum, krumme13, kinder-im-herzen) for individuals who self-report pedohebephilic interests. Informed consent was obtained at the beginning of the online survey, where participants indicated their agreement by selecting the option "I agree" from a checklist. Power analyses, using G*Power [31], indicated that 193 participants were required to demonstrate significant correlations at $r = .20$ (small to medium-sized effects) at $1 - \beta = .80$ and $\alpha = .05$. In order to conduct SEM, a sample size of at least 200 participants is typically recommended [32, 33]. Following our pre-registered plan, we excluded 30 cases from the initial sample ($N = 346$), because participants did not pass the quality checks (i.e., responded that they answered dishonestly for more than one answer, or indicated that their data should be thrown away because they have just clicked through), and a further 33 cases because participants reported a higher attraction to adults than to prepubescent or pubescent children.

The final dataset contained 283 cases. Participants' mean age was 34.16 years ($SD = 13.70$) and 249 reported "male" as their sex, while 34 reported "female". Most participants ($n = 239$)

stated to have completed at least 12 years of education. Fifty-four percent had stronger sexual attraction to prepubescent than to pubescent children, while 14% reported an equal attraction to both, and 31% were more attracted to pubescent than to prepubescent children. There were roughly equal portions of the sample who reported a stronger attraction to either male (42%) or female children (49%), with 9% reporting to be equally attracted to male and female children. Some reported prior convictions for sexual crimes involving children (19%). Among these, 4% reported previous convictions for child sexual abuse only, 12% for child pornography offenses only, and 2% for both.

## Measures

**Attitudes toward seeking treatment.**    Attitudes toward seeking treatment was assessed with the Attitudes Toward Seeking Professional Psychological Help Scale [34, 35], a self-report scale consisting of 10 items with a 4-point response scale. Factor analytic research has found support for a three-factor model with the scales *Openness to seeking professional help*, *Value in seeking professional help*, and *Preference to cope on one's own* [35]. Attitudes toward seeking professional psychological help and was found to relate to prior treatment seeking [34] and to less perceived stigma about mental health treatment [36]. When the three first-order factors are part of a higher-order factor, the resulting model is just-identified, which implies that the fit of the higher-order model should be identical to the first-order factor model. Because of this, we will use a second-order model with the construct willingness to seek treatment as the higher-order construct and the aforementioned three factors as first order constructs in the main analyses.

**Stigma processes.**    Anticipated stigma by a therapist was measured by the *Anticipated negative therapist behavior upon disclosure* scale. This scale assesses a range of possible therapist responses following disclosure of one's sexual attraction to children. It was developed for the present research and informed by research on side effects of psychotherapy [37]. Items (see S1 Table) are rated on a 7-point Likert-type response scale from *Disagree completely* (1) to *Fully agree* (7).

General anticipated stigma was measured with the six-item *Fear of rejection/concealment* subscale from the Proximal Stigmas Scale for Minor Attracted People [FoRC; 24]. Internalized stigma was measured with the *Internalizing symptoms regarding minor attraction* from the Internalized Pedonegativity Scale [IPS; 25]. These stigma scales both use a 7-point Likert scale, with the responses options *strongly disagree* (1), *disagree* (2), *somewhat disagree* (3), *neither agree nor disagree* (4), *somewhat agree* (5), *agree* (6), and *strongly agree* (7). Both subscales have adequate internal consistency ($\omega_{FoRC}$ = .80; $\alpha_{IPS}$ = .94) and have been found to be associated with a range of negative mental health outcomes like suicidality, psychological distress, loneliness [24, 25]. Items are included in S2 and S3 Tables.

**Knowledge about psychotherapy.**    Knowledge about psychotherapy was assessed using five items designed for this study. Participants responded to items on a 101-point visual analogue scale. Scalar anchors were statements reflecting a low and high level of knowledge of a specific psychotherapy topic/process (e.g., *I do not know about my rights as a patient* and *I know a lot about my rights as a patient*; see S4 Table for the full item list). The scores were rescaled from an initial range between 1 and 101 to a range between 0 and 10 (by dividing all values by 10 and subtracting 0.1) to create a scale that is more in line with the ranges of the other scales in the subsequent SEM model.

**Psychological well-being and distress.**    Psychological distress was assessed with the Brief Symptom Inventory-18 [BSI-18; 38]. The BSI-18 has three subscales, Depression, Anxiety, and Somatization, and a second-order factor (Global Severity Index) to determine general

psychological distress [39]. BSI-18 scores show high convergent validity with the longer version of the scale, the Symptom Check List-90 [40], as well as with independently developed screening scales for depression and anxiety [39]. Psychological well-being was assessed with the 5-item WHO-5 Well-being index [41]. Items are scored as follows: (5) All the time, (4) Most of the time, (3) More than half of the time, (2) Less than half of the time, (1) Some of the time, (0) At no time. Higher scores on the WHO-5 reflect higher psychological well-being. The WHO-5 has demonstrated sufficient internal consistency and test-retest reliability [42] as well as validity as a screening tool for depression and an outcome measure for clinical trials [43].

**Self-reported sexual attraction.** Erotic age orientation was assessed using attraction ratings towards people from different sex (male, female) and maturity categories (prepubescent, early-to-mid pubescent, postpubescent). Participants rated their level of sexual attraction to each on a 10-point Likert-type scale ranging from 1 (no sexual interest) to 10 (maximum sexual interest). Each item consisted of a stimulus depicting a drawn, naked character (e.g., of a pubescent girl) and a corresponding description (e.g., "How strong is your sexual attraction to girls in early to mid-puberty (i.e., girls who show some signs of physical maturity like sparse pubic hair or developing breasts)?"). All drawings were designed by the artist Anna Matheja and are used with her permission. The pictures are non-pornographic and designed to demonstrate anatomy, not to cause sexual arousal. The same drawings are used to assess sexual interests in the web-based preventive intervention Troubled Desire [44]. We used difference scores to determine 1) a Pedohebephilic Index (maximum level of sexual attraction to prepubescent or pubescent children minus maximum level of sexual attraction to adults), 2) a Male Pedohebephilic Index (maximum level of sexual attraction to male prepubescent or pubescent children minus maximum level of sexual attraction to female prepubescent or pubescent child), and 3) a Pedophilic Index (maximum level of sexual attraction to prepubescent children minus maximum level of sexual attraction to pubescent children). The measure has shown convergent validity to assess sexual interest with an implicit viewing time task [45]. The measure's test-retest reliability has not been determined.

**Sociodemographic information.** We assessed participants' sex ("male", "female"), age, and educational achievement as the highest grade in elementary school or secondary school/ high school that participants finished and got credit for.

**Quality check questions.** Please see Aust et al. [46] and Sischka et al. [47] for the exact formulation of the seriousness and the honesty check item. Participants reporting that they had answered more than one item dishonestly ($n = 21$) were excluded, as were those indicating that they have not taken part seriously ($n = 13$).

## Data analysis plan

Confirmatory factor analyses (CFA) and SEMs were conducted in the R package *lavaan* [48]. As some items showed severe deviations from the assumption of normality (as indicated by skewness outside of +/−2 or kurtosis outside of +/−7), we conducted maximum likelihood (ML) estimation with a Satorra-Bentler-corrected test statistic. RMSEA and CFI results will be reported based on Brosseau-Liard and Savalei's [49] nonnormality correction. Treating ordinal variables as continuous may lead to imprecise parameter estimates and standard errors [50]. As the scale assessing attitudes toward seeking treatment only includes four response alternatives [50, 51], we considered using *weighted least squares means and variance adjusted* (WLSMV) estimator, which is more adequate for ordinal data [52], as a sensitivity test. However, WLSMV may perform worse than ML in small samples and nonnormal latent distributions [53]. Furthermore, the use of WLSMV created identification problems in the current data, which led to the removal of two subscales from the main outcome (attitudes toward

seeking treatment). Therefore, the present study will only perform ML. The global fits of the models were assessed according to the following (*post hoc*) criteria: comparative fit index [CFI] >.90, root mean square error of approximation [RMSEA] < .08 [54]. Note that there is debate in the literature about interpreting model fit [55] and that the CFI and RMSEA cutoff criteria selected here are more liberal than what has been described as acceptable by some scholars [e.g., 56]. In models with small degrees of freedom, greater emphasis was placed on CFI than RMSEA [57].

**Stage 1. Assessing the factor structure of all new scales.** Initially, we verified that the newly developed or less established scales worked adequately well on their own, based on RMSEA and CFI criteria. Scales were eliminated or collapsed with other scales if correlations with other scales were above.80, and items with factor loadings below 0.40 were eliminated [58]. Furthermore, we considered specifying residual correlations between items (if indicated by the model fit and modification indices following standard criteria). In one case, a new scale continued to deviate substantially from the threshold of CFI = .90, which prompted us to assess violations of the assumption of local independence. To detect locally dependent (i.e., redundant) items, we employed unique variable analysis, which detects local dependency in latent variable models without a priori knowledge about the internal structure [59]. Weighted topological overlap-values above.20 for a pair of variables were used as a threshold to conclude that the assumption of local independence is violated. In these cases, we deleted one of the two variables, giving preference to items that were more general, had higher variance, and better covered the construct [59].

**Stage 2. Testing associations.** After confirming adequate fit for the measurement models of the new scales, bivariate correlations were assessed based on the latent variables that emerged from conducting separate CFAs for all included scales. SEM model were used to test all variables as part of a larger, multifactorial model, with mediation effects. If the model fit of the preregistered SEM model including all variables was below the CFI or RMSEA threshold, we picked those factors that are most reasonable from a theoretical point of view and within small range of the best CFI and RMSEA. We could not conduct the pre-registered FIML procedure, as it is not available for ML estimation. Using listwise deletion, the pre-registered (SEM) model (Model 1) was conducted with a sample size of 275 out of 283, indicating a rate of missing values of 2.8%. As this rate was considered negligible, no further missingness analyses were conducted. The pre-registered SEM model included mediation (i.e., indirect effects).

## Pre-registration, data, materials, and code

The current study design, variables, hypotheses, materials (incl. informed consent), and analyses are pre-registered as Research Question 2 of the project "Better therapy for people who are attracted to children" on Open Science Framework (https://osf.io/fd34k and https://osf.io/kpza6/), alongside two other research questions, which have already been published [60, 61]. Deviations from the preregistration are listed in S1 File. The raw data can be requested from the Norwegian Agency for Shared Services in Education and Research (Sikt) via https://doi.org/10.18712/NSD-NSD3178-V2. Note that the dataset is available for researchers only, in line with the information provided in our consent procedure, to increase participants' expected anonymity. The syntax is available without restrictions via the same link.

## Results

## Stage 1. Assessing the factor structure of new scales

Factor loadings for all CFAs, as well as means and standard deviations of individual items are presented in S1–S4 Tables.

**Anticipated negative therapist behavior upon disclosure.**   The expected unifactorial model deviated substantially from CFI >.90 and RMSEA < .08, $\chi^2$ = 526.78, $df$ = 104, $p$ < .001, CFI = .82, RMSEA = .15, 95% CI [.14,.16]. Unique variable analysis indicated that the assumption of local independence was violated for five variable pairs. In all cases, the redundant items were likely to arise from shared item phrasing (i.e., "respect patient confidentiality" and "treat me with respect", "pay close attention to my needs" and "choose therapeutic interventions that suit my goals and needs") and/or highly similar item content (e.g., "stop seeing me" and "tell me that he or she cannot work with me any longer", "report me to the authorities" and "warn others about me", "pressure me to do interventions that I do not want to do" and "try to change my sexual interests"). Therefore, we removed one item from each pair. An analysis of the reduced 11-item scale did not yield a satisfactory model fit, $\chi^2$ = 123.38, $df$ = 44, $p$ < .001, CFI = .94, RMSEA = .10, 95% CI [.08,.12]. We achieved the best fit for a three-factor second-order model with one higher order factor (see S2 Table), $\chi^2$ = 77.16, $df$ = 41, $p$ < .001, CFI = .97, RMSEA = .07, [.05,.09]. The three factors are (1) *Reduced Collaboration and Respect*, characterized by an anticipated decrease in collaboration around treatment goals and respect shown by the therapist, (2) *Adverse Therapist Behavior*, characterized by anticipating therapist acting in ways that are harmful, and (3) *Treatment Discontinuation*, characterized by anticipating a therapist to end treatment.

**General anticipated stigma.**   For the FoRC, the last item ("At school and/or work, I pretend to be interested in adults [e.g., pretending to be attracted to women or men]") did not account for sufficient common variance (standardized factor loading = .23) and was removed from the model. The model fit of the scale without the last item was acceptable, $\chi^2$ = 3.64, $df$ = 5, $p$ = .602, CFI = 1.00, RMSEA = .00, 95% CI [.00,.10].

**Internalized stigma.**   CFA revealed unacceptable fit of the subscale, $\chi^2$ = 68.57, $df$ = 20, $p$ < .001, CFI = .94, RMSEA = .10, [.08,.13]. We deleted the item "Most MAPs end up lonely and isolated," because of its low standardized factor loading (.36). The fit of the model was acceptable, $\chi^2$ = 42.26, $df$ = 14, $p$ < .001, CFI = .96, RMSEA = .10, [.06,.13].

**Knowledge about psychotherapy.**   CFA supported the expected unifactorial structure of the scale, $\chi^2$ = 8.23, $df$ = 5, $p$ = .144, CFI = .99, RMSEA = .06, 95% CI [.00,.14]).

## Stage 2. Testing associations

**Bivariate associations.**   The correlation matrix, as well as means and standard deviations of the average composite scores of all scales can be found in Table 1. Out of 11 pre-registered hypothesized associations, nine were statistically confirmed in our dataset. In line with our expectations, we detected zero-order associations between attitudes toward seeking treatment on the one hand and knowledge about psychotherapy, anticipated negative therapist behavior upon disclosure, internalized stigma, and psychological distress on the other hand. Against our expectations, no statistical associations were found between general anticipated stigma or psychological wellbeing on the one hand and attitudes toward seeking treatment on the other hand. For psychological distress, we corroborated the expected link to higher internalized stigma, but not to general anticipated stigma. Furthermore, we detected a link between psychological wellbeing and lower general anticipated stigma, as well as lower internalized stigma. The hypothesized negative link between anticipated therapist maltreatment upon disclosure and knowledge about psychotherapy was also confirmed.

**SEM models.**   The CFI of the pre-registered model was below the threshold of .90, $\chi^2$ = 26447.24, $df$ = 1743, $p$ < .001, CFI = .89, RMSEA = .05, 95% CI [.04,.05]. Psychological wellbeing *positively* predicted attitudes toward seeking treatment in the pre-registered SEM model ($\beta$ = .34, [.05,.32]). This indicates a suppression effect, as psychological wellbeing was not

**Table 1. Mean and standard deviations for average composite scores and correlations based on latent variables.**

| Variable Name | Average composite scores | | Correlation matrix based on latent scores from CFAs (binary associations) | | | | | |
|---|---|---|---|---|---|---|---|---|
| | *M* | *SD* | 1 | 2 | 3 | 4 | 5 | 6 |
| 1. Attitudes toward seeking treatment | 2.84 | 0.66 | | | | | | |
| 2. Psychological wellbeing | 2.41 | 1.12 | -.05 | | | | | |
| 3. Psychological distress | 2.08 | 0.78 | .20** | -.58** | | | | |
| 4. Knowledge about psychotherapy | 6.02 | 2.56 | .30** | -.03 | .11 | | | |
| 5. Anticipated negative therapist behavior upon disclosure | 4.05 | 1.48 | -.27** | -.13* | .05 | -.22** | | |
| 6. General anticipated stigma | 5.90 | 1.30 | -.02 | -.13* | .09 | -.20** | .19** | |
| 7. Internalized stigma | 3.70 | 1.59 | .26** | -.43** | .41** | -.05 | .16* | .44** |

* indicates *p* < .05.

** indicates *p* < .01.

The average composite scores range from 1 to 4 for attitudes toward seeking treatment, from 0 to 5 for Psychological Wellbeing, from 1–5 for Psychological distress, from 0–10 for Knowledge about psychotherapy, and from 1–7 for Anticipated negative therapist behavior upon disclosure, General anticipated stigma, and Internalized stigma.

significantly related to attitudes toward seeking treatment in bivariate analyses (*r* = -.05, see Table 1). It was our initial intention to include the WHO-5 alongside the BSI-18 in order to include in our model constructs that are not solely from a deficit/disorder perspective. In hindsight, the strong conceptual overlap of these two measures and constructs was likely foreseeable as a potential problem and likely represents a theoretical error in model building on the part of the authors. It was therefore decided to remove psychological well-being from the model to increase interpretability and to account for the strong conceptual overlap between psychological wellbeing and psychological distress.

After removing psychological well-being from the model, the fit for the modified SEM model (Model 2, see Fig 1 for an overview including standardized regression coefficients) was as follows, $\chi^2$ = 2170.61, *df* = 1462, *p* < .001, CFI = .905, RMSEA = .04, [.04,.05]. More positive attitudes toward seeking treatment were associated with higher internalized stigma, lower anticipated negative therapist behavior upon disclosure and higher knowledge about psychotherapy, while showing no significant link to general anticipated stigma and psychological distress. Direct, indirect, and total effects of the mediation hypotheses are shown in Table 2 for both models. Anticipated negative therapist behavior upon disclosure partially mediated the link between knowledge about psychotherapy and attitudes toward seeking treatment. There were no significant indirect effects between internalized stigma or general anticipated stigma and attitudes toward seeking treatment via psychological distress. Of note, we discovered a small negative association between general anticipated stigma and psychological distress in both Model 1 and 2, even though the bivariate association between the two constructs was nonsignificant and in the opposite direction. This unexpected negative association may reflect a protective effect of general anticipated stigma, that only emerges when stripped of its overlap with internalized stigma. A comprehensive overview of all unstandardized model parameters for Model 1 and 2 can be found in the S5 and S6 Tables.

Out of 1540 residual correlations, the absolute values (modulus) of 263 were between .10 and .20, 19 between .20 and .30, and none was larger than .30. Some items stood out as particularly problematic. The item "pressure me to do interventions that I do not want to do" had residual correlations ≤−.10 with six items from the Internalized Stigma scale, particularly three items (residual correlations ≤−.20) assessing positive feelings towards one's sexual

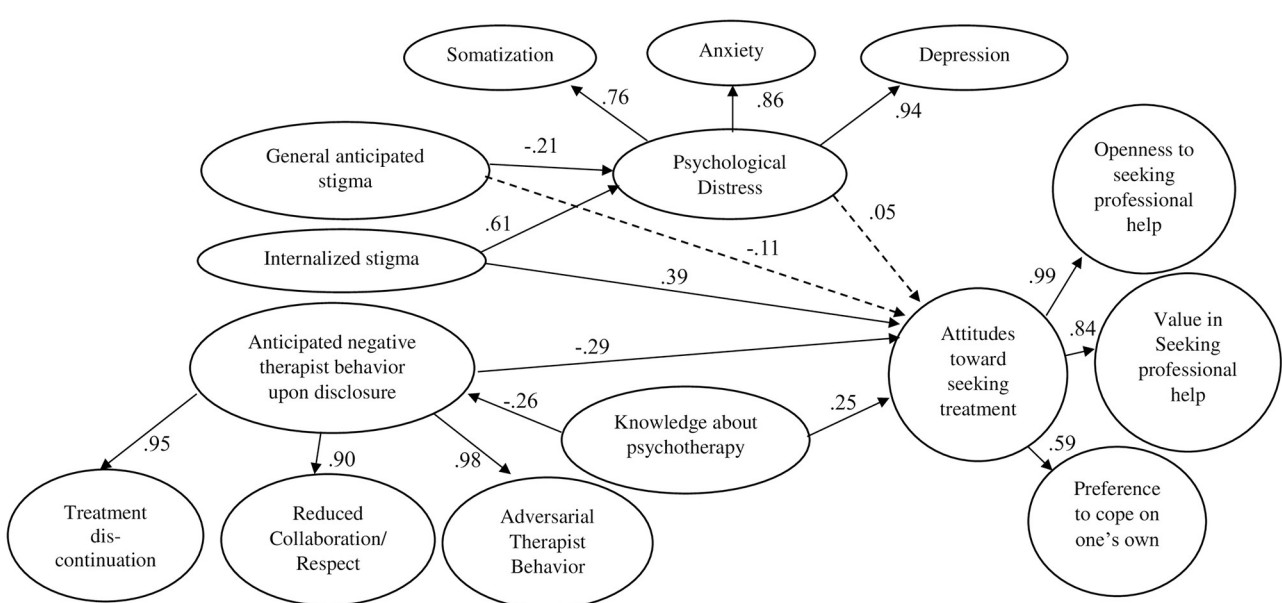

**Fig 1. Standardized regression coefficients for Model 2.** Items are not depicted to increase readability. Second-order factors are depicted with their respective first order factors. Dashed lines are not significant ($p < .05$). Covariances: General anticipated stigma and internalized stigma: β = .51 ($p < .001$), General anticipated stigma and anticipated negative therapist behavior upon disclosure: β = .15 ($p = .016$), Internalized stigma and anticipated negative therapist behavior upon disclosure: β = .18 ($p = .005$), Knowledge about psychotherapy and Internalized stigma: β = -.05 ($p = .516$), General anticipated stigma and knowledge about psychotherapy: β = -.24 ($p = .001$).

attraction (note that these items were inverted, such that higher scores represent more internalized stigma). The item "Whenever I think about being an MAP, I feel depressed" showed residual correlations ≥.10 with all items from the BSI-18 Depression subscale. We attempted two model specifications to account for this problem, 1) conducting the analyses without item 4, 2) including residual covariances between the item and the Depression subscale, changing the model fit to $\chi^2 = 2039.41$, $df = 1408$, $p < .001$, CFI = .91, RMSEA = .04, [.04,.05] and $\chi^2 = 2163.44$, $df = 1461$, $p < .001$, CFI = .91, RMSEA = .04, [.04,.05], respectively. As the changes in

**Table 2. Parameter estimates for direct, indirect, and total effects for Model 1 and 2.**

| Hypothesized Mediation Effect | | Model 1 | | | | Model 2 | | | |
|---|---|---|---|---|---|---|---|---|---|
| | | Std. estimate | Unstandardized estimate (SE) | lower limit 95% CI | upper limit 95% CI | Std. estimate | Unstandardized estimate (SE) | lower limit 95% CI | upper limit 95% CI |
| Internalized stigma → Psychological distress[1] → Attitudes toward seeking treatment | Direct | .42 | 0.20 | 0.10 | 0.29 | .39 | 0.18 | 0.09 | 0.28 |
| | Indirect | .00 | 0.00 | -0.05 | 0.05 | .03 | 0.02 | -0.03 | 0.06 |
| | Total | .42 | 0.20 | 0.12 | 0.27 | .42 | 0.20 | 0.12 | 0.28 |
| General anticipated stigma → Psychological distress[1] → Attitudes toward seeking treatment | Direct | -.12 | -0.06 | -0.14 | 0.02 | -.11 | -0.06 | -0.14 | 0.02 |
| | Indirect | .00 | 0.00 | -0.03 | 0.02 | -.01 | -0.01 | -0.02 | 0.01 |
| | Total | -.12 | -0.06 | -0.14 | 0.02 | -.12 | -0.06 | -0.14 | 0.02 |
| Knowledge about psychotherapy → Anticipated negative therapist behavior upon disclosure → Attitudes toward seeking treatment | Direct | .26 | 0.07 | 0.03 | 0.11 | .25 | 0.07 | 0.03 | 0.11 |
| | Indirect | .07 | 0.02 | 0.01 | 0.04 | .08 | 0.02 | 0.01 | 0.04 |
| | Total | .33 | 0.09 | 0.05 | 0.13 | .33 | 0.09 | 0.05 | 0.14 |

[1] Note that in Model 1, psychological wellbeing was included as a second mediator alongside psychological distress.

model fit compared to Model 2 were minimal, neither specification was adopted. There was no theoretical legitimization to respecify the model further.

## Discussion

The present research examined the association between different stigma processes, knowledge about psychotherapy, psychological distress, and attitudes toward seeking treatment in pedohebephilic individuals. Our analyses indicated differential associations for different stigma processes. Anticipated negative therapist behavior upon disclosure was linked to less positive attitudes toward seeking treatment, while internalized stigma was linked to more positive attitudes toward seeking treatment, and no association was detected for general anticipated stigma. In line with previous evidence [e.g., 15], knowledge about psychotherapy was linked to more positive attitudes toward seeking treatment. Anticipated stigma was only related to attitudes toward seeking treatment when it was assessed in relation to therapists. This likely explains why past research has not detected significant links between general anticipated stigma (in non-specified social situations) and attitudes toward seeking treatment [12]. Our study also highlights that some stigma processes (i.e., internalized stigma) can relate to more positive attitudes toward seeking treatment. Therefore, it is crucial for future research interested in the link between stigma and attitudes or behavioral intentions to assess multiple components of stigma, as they may be differentially related to these attitudes or intentions.

In research on mental illness stigma, higher internalized stigma is typically associated with a less positive attitudes toward seeking treatment [8, 9]. However, in our present sample, as well as a small number of previous studies on the stigma associated with pedophebephilia [14], there was a positive relationships between internalized stigma and attitudes toward seeking treatment. This raises questions regarding possible explanations for this differential influence of internalized stigma in different populations. A reasonable explanation seems to be that the more pedohebephilic individuals have adverse reactions about their attraction, the more likely they are to want treatment to alleviate their feelings of confusion, shame, or self-hatred. Hence, type of stigmatized group (e.g., pedohebephilia vs. mental illness like depression or schizophrenia) may moderate the association between internalized stigma and treatment seeking.

In line with expectations from the minority stress model [19] and previous research [25], internalized stigma was associated with more psychological distress. For general anticipated stigma, higher anticipated stigma was associated with less psychological distress and higher wellbeing in a SEM model. This finding hints at a potential mediating effect, as anticipated stigma might activate coping strategies to manage situations where the stigmatizing attribute might be discovered. We assume that this association, which was not detectable in bivariate analysis, could only emerge because internalized stigma absorbed much of the shared variance related to psychological distress. However, note that caution is also warranted when interpreting unexpected findings, and we want to stress that more research is needed to establish whether this result replicates.

Finding differential associations for various stigma processes with attitudes toward seeking treatment has important implications for interventions to increase treatment seeking. Anticipating negative therapist behavior upon disclosure was associated with less knowledge of psychotherapy, suggesting that this stigma process is informed to a certain extent by a general unfamiliarity with mental health services. The accuracy of these beliefs is difficult to ascertain, though recent research offers some insight. By all accounts, extreme stigmatization of pedophilic clients appears to be less common among therapists than the general public [62]. Nevertheless, 40% of Swiss therapists agreed that all pedohebephilic individuals should receive

mandated treatment and one in five believed that pedohebephilic individuals are destined to commit child sexual abuse [62]. There is also evidence that stigmatizing attitudes can inform decision-making by clinicians, including making mandated reports in situations with pedohebephilic clients that do not warrant a report [63] or their willingness to offer services to this client population [64]. Further, there is a certain level of disagreement in the goals of treatment between nonclinical samples of pedohebephilic individuals and mental health professionals [6, 65]. From the available evidence in these related lines of research, it seems plausible that while anticipatory stigma is a barrier to treatment, not seeking treatment can serve a protective function if one expects rejection by or negative consequences from working with a therapist.

While online mental health literacy interventions can increase knowledge about psychotherapy, a previous meta-analysis indicates that they may be limited in their ability to increase positive attitudes toward seeking treatment [66, 67]. Hence, we would suggest an intervention that combines education about pedohebephilia with interventions that challenge the assumptions that therapists who would be willing and capable to provide high quality care do not exist. Previous research has shown that intergroup contact interventions, even when delivered indirectly via video, can improve outgroup attitudes and behaviors [e.g., 68]. An intervention to increase trust in therapists could include videos or stories involving a warm, non-judgmental therapist. As a caveat, such an intervention would also need to acknowledge that some clients experience rejection, hostility, or abandonment upon disclosing their sexually attracted to children to their therapist [69]. To mitigate these risks, an intervention could provide the clients with strategies to describe their sexual interest in ways that increase their chances of receiving the support that they need, for instance by communicating that having sexual interests in children and committing sexual offenses are not the same. However, such an intervention component would have to designed carefully, as clients who are at risk of committing sexual offenses should not be encouraged to conceal or downplay risk-relevant factors in treatment. Furthermore, potential clients should be encouraged to seek treatment with therapists that are specialized in working with this client group, if possible.

In line with expectations from the minority stress model [19] and previous research [25], internalized stigma was associated with more psychological distress in the present sample. For general anticipated stigma, this study indicates a more complex relationship, where higher anticipated stigma was associated with less psychological distress and higher wellbeing in a SEM model. This finding hints at a potential mediating effect, as anticipated stigma might activate coping strategies to manage situations where the stigmatized attribute might be discovered. We assume that this association, which was not detectable in bivariate analysis, could only emerge because internalized stigma absorbed much of the shared variance related to psychological distress in the SEM model. However, caution is warranted when interpreting unexpected findings, and we want to stress that more research is needed to establish whether this result replicates.

Lastly, our results have implications for clinical work. The relationship between internalized stigma and attitudes toward seeking treatment suggests that when entering treatment, pedohebephilic clients are experiencing elevated levels of shame and distress around their sexual attractions. Recent research on a subset of the current sample rounds out this set of findings, suggesting that a positive therapeutic relationship and trust in a therapist can influence whether pedophebephilic clients disclose their sexual attractions in therapy [61]. Communication, for instance on a professional or clinic website, conveying openness and competence to treat individuals with pedohebephilic interests in a non-judgmental and ethical manner may have value for this client population.

## Limitations

The foregoing implications need to be tempered by the limitations of the present research. The present study focuses strongly on stigma and attitudes toward seeking treatment as a direct connection, excluding potential mediators (with the exception of psychological distress). Future studies should consider other potential mediators like general self-devaluation. Correlational studies in general are limited in that they can identify an association but not causation. Rather than internalized stigma *causing* mental disorders, it is also possible that mental disorders increase the likelihood of internalizing stigma or that both are affected by a third variable. Bailey [70] discusses rival explanations for a link between psychological distress and stigma in LGB populations, which may also be relevant for pedohebephilic clients. These problems may be alleviated by using experimental (e.g., via testing the effects of de-stigmatizing messages on mental health and treatment motivation) or longitudinal designs. Due to their reach and perceived anonymity, online surveys are the only viable option to survey a large sample of pedohebephilic individuals from the community. Yet, these sampling strategies are also at a high risk for self-selection bias, and our survey is likely to oversample Western participants with a high mental health literacy, high fluency in English, and strong interest in improving treatment options for pedohebephilic people.

For several scales the proposed models did not fit our data as well as expected. It was possible to increase fit based on modifications, which were theoretically plausible, and, with the exception of the anticipated therapist behavior upon disclosure scale, minor. Nevertheless, we acknowledge that this practice can change the meaning of the scale in unforeseeable ways. A problem with the attitudes toward seeking professional psychological help scale is that it specifically pertains to seeking professional help in response to emotional problems. Although previous studies show that such concerns are common among pedohebephilic people who seek treatment [65, 69], other motives for treatment (such as seeking validation or coping with sexuality) are not covered explicitly by this scale. Moreover, future studies might consider assessing interest in counseling or support that do not require formal psychotherapeutic training and mental health professional status. Although we achieved a good fit for a three-factor higher-order structure for anticipated therapist behavior upon disclosure, concerns about potential overfitting warrant caution. Further research is needed to validate and possibly refine the factor structure of this scale to ensure its reliability and validity in future studies.

## Conclusions

The findings highlight the importance of studying stigma processes specific to pedohebephilic populations and therapy itself when understanding treatment seeking in this population. Of most interest, we found that internalized stigma may increase willingness for treatment, while anticipated stigma was only negatively linked to attitudes toward seeking treatment when it was assessed in relation to the therapeutic context. This clarity may help to inform both public messaging campaigns regarding psychotherapy for pedohebephilic clients and clinicians informing pedohebephilic clients of the processes of therapy, consent, and reporting obligations.

## Supporting information

**S1 Table. Means, SDs and standardized factor loadings of items from the anticipated negative therapist behavior upon disclosure scale (N = 286).**
(DOCX)

**S2 Table. Means, SDs and standardized factor loadings of items assessing anticipated stigma (from fear of rejection/concealment subscale of the proximal stigmas scale for minor attracted people, N = 283).**
(DOCX)

**S3 Table. Means, SDs and standardized factor loadings of items for internalized stigma (from the internalizing symptoms regarding minor-attraction subscale, N = 283).**
(DOCX)

**S4 Table. Means, SDs and standardized factor loadings of items from the knowledge about psychotherapy scale (N = 286).**
(DOCX)

**S5 Table. Model parameters for Model 1 (pre-registered model).**
(DOCX)

**S6 Table. Model parameters for Model 2.**
(DOCX)

**S1 File. Deviations from the preregistration.**
(DOCX)

**S1 Checklist. Human participants research checklist.**
(DOCX)

## Author Contributions

**Conceptualization:** Sara Jahnke, Ian V. McPhail, Jan Antfolk.

**Data curation:** Sara Jahnke.

**Formal analysis:** Sara Jahnke, Ian V. McPhail, Jan Antfolk.

**Funding acquisition:** Sara Jahnke, Jan Antfolk.

**Investigation:** Sara Jahnke.

**Methodology:** Sara Jahnke, Ian V. McPhail.

**Project administration:** Sara Jahnke.

**Writing – original draft:** Sara Jahnke.

**Writing – review & editing:** Sara Jahnke, Ian V. McPhail, Jan Antfolk.

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
