## [Decision Letter · Decision Letter 0]

3 Apr 2024

PONE-D-24-03768Stigma processes and willingness to seek treatment among pedohebephilic peoplePLOS ONE

Dear Dr. Jahnke,

Thank you for submitting your manuscript to PLOS ONE. After careful consideration, we feel that it has merit but does not fully meet PLOS ONE’s publication criteria as it currently stands. Therefore, we invite you to submit a revised version of the manuscript that addresses the points raised during the review process.

**I have received three excellent and thorough reviews for your manuscript, two suggesting major revisions and 1 suggesting minor revisions. Reviewer 1 emphasized the need for a more detailed discussion to enhance its depth. Reviewer 2 suggested more comprehensive methodological revisions to ensure thoroughness. Additionally, Reviewer 3 recommended further clarification of the analytic protocol, pinpointing areas for improvement. With these precise and constructive reviews, you can refine your manuscript effectively, ensuring clarity and rigor in your research before acceptance.**

We look forward to receiving your revised manuscript.

Kind regards,

Inga Schalinski

Academic Editor

PLOS ONE

Journal Requirements:

This study was funded by a personal grant (application number 333761) to the first author from the Academy of Finland.

4. In the online submission form you indicate that your data is not available for proprietary reasons and have provided a contact point for accessing this data. Please note that your current contact point is a co-author on this manuscript. According to our Data Policy, the contact point must not be an author on the manuscript and must be an institutional contact, ideally not an individual. Please revise your data statement to a non-author institutional point of contact, such as a data access or ethics committee, and send this to us via return email. Please also include contact information for the third party organization, and please include the full citation of where the data can be found.

5. Thank you for uploading your study's underlying data set. Unfortunately, the repository you have noted in your Data Availability statement does not qualify as an acceptable data repository according to PLOS's standards.

6. We note that you have referenced McPhail IV, Stephens S. which has currently not yet been accepted for publication. Please remove this from your References and amend this to state in the body of your manuscript: (McPhail IV, Stephens S. [Submitted]) as detailed online in our guide for authors

7. We note that you have referenced Chronos A, Jahnke S, Blagden N. which has currently not yet been accepted for publication. Please remove this from your References and amend this to state in the body of your manuscript: (Chronos A, Jahnke S, Blagden N. [Submitted]) as detailed online in our guide for authors

Reviewers' comments:

Reviewer's Responses to Questions

**Comments to the Author**

1. Is the manuscript technically sound, and do the data support the conclusions?

Reviewer #1: Yes

Reviewer #2: Yes

Reviewer #3: Partly

2. Has the statistical analysis been performed appropriately and rigorously? 

Reviewer #1: Yes

Reviewer #2: No

Reviewer #3: Yes

3. Have the authors made all data underlying the findings in their manuscript fully available?

Reviewer #1: Yes

Reviewer #2: Yes

Reviewer #3: No

4. Is the manuscript presented in an intelligible fashion and written in standard English?

Reviewer #1: Yes

Reviewer #2: Yes

Reviewer #3: Yes

5. Review Comments to the Author

**Reviewer #1:** I enjoyed your paper.

The main problem I found is that there are two important, recent studies on this topic that are ignored in the article: Roche and Stephens, 2022 (“Clinician stigma and willingness to treat those with sexual interest in children”), and Schmidt and Niehaus, 2022 (“Outpatient Therapists’ Perspectives on Working with Persons Who Are Sexually Interested in Minors”). Once we consider findings from these two studies, your claim that “extreme stigmatization of pedophilic clients appears to be rare among therapists” should be much more nuanced, and your discussion (starting at line 477) about the accuracy of pedophebephilic individuals’ beliefs and anticipations of negative attitudes from therapists should be different.

Roche and Stephens, 2022 found that 48% of therapists from their sample did not want to work with pedophebephilic individuals who had a past offense and wanted help managing their attraction. Schmidt and Niehaus, 2022 found that 20% of their Swiss therapists believed that pedophebephilic individuals were destined to sexually abuse a child; 40% said that ALL pedophebephilic individuals should be mandated to receive psychotherapy, and 57% agreed with a statement saying that people who are attracted to children are “sick”.

Some of these findings need to be mentioned, and your interpretation in the discussion section needs to be modified accordingly. Similarly, your recommendation to develop education campaigns aiming to “decrease fears of negative therapist behavior” among pedophebephilic individuals seems premature in light of these other findings. Such education campaigns are first and foremost needed to change therapists’ views about pedophebephilic individuals.

I have other more minor points.

On lines 97 and 98, you write: “Studies on the link between anticipating rejection and seeking treatment in pedophebephilic individuals have produced mixed results (8).” The reference to (8) here seems to be wrong. It links to Clement et al. (2015), which does not discuss pedophebephilic individuals.

In relation to the previous point on anticipating rejection: I am surprised that there is no mention of the often-cited B4U-ACT 2011 survey results here. In that survey, the top reason given for MAPs not seeking professional help is: "Fear of negative reaction by the professional (78%)".

Finally, in various places, the expression “sexual minorities” is used, and its use seems to include LGB populations but exclude pedophebephilic individuals (example of exclusion on line 475: "(e.g.,sexual minority or pedohebephilia)”. It’s not clear at all why pedophebephilic individuals wouldn’t count as a “sexual minority", and you

never define the expression. To avoid confusion, you should either 1) drop the expression “sexual minority” and stick to LGB or 2) drop the opposite between pedophebephilic individuals and sexual minorities.

**Reviewer #2:** The current article investigates a number of important scales with respect to the motivation of pedohebephilic patients to seek treatment. This is a very important topic, and it is of great importance that the authors work on scales that are crucial for other authors to research this topic. The authors on top of that present an interesting mediation result that may explain differing previous results in the literature. There are a number of issues with the methodology used or the presentation of the methodology, at some points I admit I’m not sure which of the two. To make sure that the important results are not hidden behind these problems, I think the manuscript would benefit from some clarification here before publication. The only major change I would suggest is to remove the WLSMV estimator from the article since it is theoretically less founded, created some practical problems with this data set which forced to remove two subscales, and other detail reasons I outline below.

Major Issues:

line 217-220: The scale that participants were able to answer from is missing. It seems from the means in Table S7 that items were on a scale from 1 to 100, but that doesn’t seem to match a total score from 1 to 10. Computing ordinal scores that correspond to ranges suggest a five-point Likert scale. You may want to clarify.

line 297: Which missingness treatment strategy did you use instead of FIML for the remaining 2.8% missingness? It seems from the reduction of sample size that you used listwise deletion; note that this is introducing a bias (although admittedly probably small for only eight participants). In doubt, I guess FIML on uncorrected data ML is the better choice; you may want to simulate to be sure (or just do both and compare results, if the outcome is more or less the same anyway, who cares).

line 316: A fit of CFI = .97 for a factor model with five items is fully acceptable, the RMSEA is less useful for such a model. Adding it is reducing the dfs from 5 to 4, there’s not much space left to perfect fit. it increases overfitting, and it makes the interpretation difficult (since the factor scores can no longer be computed as a weighted sum of the item scores). I’d suggest to just go with the original model.

factor scale improvements: You seemed to choose very different strategies for improving the factor models, removing items, adding residual covariances, or changing the number of factors. All methods are absolutely fine, but you run in the danger of overfitting when applying many of those strategies, a little more so if you don’t give a fixed strategy in which order you try them and when already reacting on fairly good fit values just if they are a little bit over (although that is often not acknowledged) arbitrary cut-off values. My suggestion would be to fix a strategy, here would be the one I’d suggest: For all reasonable number of factors, delete all items with a standardized loading below a given threshold (or whenever CFI is improving, that is fair too), and then pick the number of factors that are most reasonable from a substantive point of view and within small range of the best CFI. A purist might say it’s still overfitting (and that is probably true), but it would drastically reduce that effect.

line 321: You later report the three factors of the BSI as (I think) Somatization, Depression, and Anxiety; if I got that correct, I suggest to introduce the names here.

Table 1 Footnote a: Why did you use average scores for the means suddenly? That seems surprising, considering that your factor weights are sometimes a good deal away from uniform distribution. Also (this very minor) a combined score from manifests is usually no longer called a manifest variable, it’s a computation of a latent.

Table 1 correlations: Why did you use the latent scores to compute the correlations? Wouldn’t it make more sense to fix the factor loadings and use the latent covariances in an SEM that includes all factors? Note that factor scores are good for means, but not so much for everything to do with variances. However, in case there is a strong reason for that, better than nothing certainly. Finally, you indicate the table contains “binary demographic variables”, I must have missed something here, aren’t these nine all latent factors?

Table 1 Footnote b: I may have missed that before, but this is the only time that you say that you used a second-order factor model, correct? You may want to clarify in Line 322-329 (and, if applicable, the other factors) whether the CFI reported there is from a two/three factor model with independent factors, with a second-order superfactor or maybe with correlated factors. I also assume that the comment regarding treating the variables as continuous holds for the whole table and not just “Willingness to Seek Treatment”? If so, you may want to put it in the notes (although honestly, I also think you can omit that, it makes sense here anyway).

Table 1 Variable 7: You report earlier you used a three-factor model for this construct (even gave the names of the three factors), why and how (second order factor model maybe?) do you combine them here (but leave 3, 4, and 5 separated)?

line 368-369: The beta is negative, why do you say it predicted positively?

line 369-370: A null result in NHST does not imply that the variables were unrelated (see remark regarding stars above, it seems you may have been affected by this).

line 369-370: I may miss that, but what variable do you think would be suppressing the correlation? Are you investigating this further or is this more of a side remark (considering that you then remove the predictor, I guess the latter, but I am a little confused here what you are aiming at).

Model 1 / Model 2: You state that Model 1 had a fit you didn’t find satisfactory, Model 2 seems to have almost exactly the same fit (the CFI is very close, and it is not a very good fit index for such a model anyway since the independent model doesn’t make much sense as a lower cut-off, and the RMSEA is exactly identical). Do you feel Model 2 is an improvement? If so, I’m not sure I understand the reason.

Table 2: Considering that both fit methods ended up with qualitatively mostly the same results, ML is more conservative in the results, ML is the one better grounded in theory, and the WLSMV fit method gave you convergence trouble with the “Willingness to seek treatment” variable so that you had to resort to Openess only (which I think is not the same thing substantially), I suggest to remove the WLSMV fit method completely from the article.

line 430-431: I do see why “Whenever I think about being an MAP, I feel depressed” and the depression subscale is correlated for MAP, this is a very hard theoretical reason to respecify the model from my point of view, contrary to your statement. It is legitimate to keep the model based on a spuriousity argument, I suppose, but I’d suggest to change the reasoning here.

line 437: By “adequate” you mean from a substantial/theoretical point of view? I don’t share that opinion, for me the other two subscales do contribute in a meaningful way. But it’s an opinion thing, so if you want to report just openess, okay. I’d suggest to generally remove the factor then and just go with openess everywhere. As mentioned above, I more strongly suggest to use the ML estimator.

Table S10: I’ve difficulties to follow the logic here: Firstly, am I right that the column header for Column 2 should go in the column itself (that is, that only the first page are factor loadings, then this column is reflecting regression slopes, and so on)? Second, why do you report all fixed-to-zero intercepts? Wouldn’t it be enough to say in the note “all intercepts were fixed to zero as all variables were moved to zero mean”? Thirdly, why do you fix all residual variances, and how did you get at the numbers you fixed them to?

line 446-448: From the CIs in Table 2 (assuming the lower and upper bounds are bounds of a CI, you may want to mention that in the table), I take it the ML estimator mostly also reported as-good-as-significant effects here, right? In a Bayesian expression of these results, the difference would still be minor. Also, I’m curious what the ML model would do if you only used openess instead of willingness – it’s really difficult to compare the two approaches on different variables. All in all, I also think from these result that you may want to take the WLSMV estimator out.

line 498-499: I’m careful here. It is a fact that visiting a psychotherapist is a substantial risk for pedohebephilia patients. I’d say convincing them by good advertisement is not the way to go as long as we cannot ascertain that the risk is in fact in correspondence to this representation of therapy. Note that I do think the majority of psychotherapists are well-intentioned, warm and successful (as most pedohebephilic persons are), but we do need to keep in mind the danger that comes from the very few exceptions.

Minor Issues (to use or not as you think best):

line 99 following: I’m not sure I understand these three sentences, do you mean the following: “A general anticipation of negative reactions, e.g., fearing to be verbally attacked in the street, did not show a reliable relation to willingness to seek treatment. It is likely that the connection is more prominent when comparing anticipation of negative reactions from a psychotherapist when confining in them.” If not, I’m sorry; you may want to define briefly what you mean by “general anticipation” and “degree of specificity”. Am I right that you assume the relation did not show because of a floor effect (because you say before that pedohebephilic persons are generally not very afraid of being confronted in the public)?

line 115 “..., with one study...” -> “... in one study...” (phrasing it less generalizable from a single study).

line 117 “This indicates...”: Seems a little strong for me to conclude this from one study looking specifically at LGB? Maybe you can rephrase it as a new hypothesis, “This makes us speculate that...” or “This leads to the possible speculation that...”

line 120 “...to want treatment..” -> “..wanting treatment...”

line 127 “...approximately half...” -> “...approximately half of the sample...”

line 96 and 110: You may want to synchronize both headers (e.g., “[dimension] Stigma and Treatment Seeking” in both. Since you identify three dimensions before, one is kind of missing “experienced stigma”. I understand you say before that experienced stigma is rare in this community compared to other stigmata, but it still exists for mostly all pedohebephilic people (in rare but memorable ways), and it seems reasonable to assume that it does have an effect, even if more difficult to measure.

below line 66: I would suggest to give an overview of possible mechanisms about the problem of the interaction to lead readers in, e.g.: “The interaction between stigma and willingness to seek therapy might go in both directions, (1) people that fear therapy will not be effective or even hurtful because of stigma anticipated from the therapist or internalized stigma will be pushed away from seeking therapy, while (2) people how anticipate, experience or internalize stigma may suffer from that fact and see an increased need for therapy for themselves. Both mechanisms may also be active at the same time and cancel out effects. The literature reflects this in reports that may seem contradictory.”

line 146, very minor: It is no longer necessarily assumed true that cross-sectional data allows no causal inference, but it is of course correct that the current data does not. “prediction” is not per se linked to “causal” either.

line 148: You may want to lead with “We hypothesize the following interactions:”

line 201: “S1” -> “S4”

supplement Table S4 notes: “UM” doesn’t appear as header, I guess you mean “UDM”

supplement Table S6 notes: “... a covariance...” -> “...an additional covariance of residuals...” (covariances exist between all items via the factor).

line 217 “S2” -> “S7”

line 234 following: You may want to indicate more precisely how you compute the scores.

line 279 “continues” -> “continuous”

line 281 “... which may be...” -> “...which are...”

Table 1 stars: I know it’s common, but among experts the stars are really frowned at, since they over-interpret post-fact NHST results and may even suggest that a lack of stars indicates a negligible correlation. My suggestion would be to mark (e.g., by making them bold) correlations you consider important substantively (e.g., that we get high correlations between variables 2, 3, 4, and 5 is not really surprising considering they come from very related constructs, three of them of the same scale).

Figure 1: The labels in the lower left three variables were cut off. Some variables are capitalized (“Psychological Distress”), other not (“Willingness to seek treatment”)

Figure 1: I’d suggest to exchange “Anticipated negative therapist behavior upon disclosure” and “Knowledge about psychotherapy” in the diagram to make the two mediation models more parallel.

Line 373: I gather that the mediation effects are taken from Model 2? If so, you may want to clarify.

Table 2 Footnote a: I guess “Seeking Professional Help”, “Attitudes Toward Seeking Professional Help Scale” and “Willingness to seek treatment” all indicate the same thing here? If so, you may want to stay with the same name, if not (sorry), you may want to clarify.

line 423: “...247 were between |.10| and...” -> “...247 had absolute values between .10 and ...” (it doesn’t make sense to make absolute symbols around the .10 since it’s a positive value anyway).

line 425: You may want to make sure that the line break is not within the number (-.10).

line 432 “Sensitivity tests...” -> “Sensitivity Tests...” (I think you want this to be a capitalized header since you capitalized elsewhere, right?)

line 435 and other places: I think items should be capitalized; I’d be happy either way, but at the moment you do it differently at multiple places.

line 436 “... Help because...” -> “...Help, because...”

line 461-462 “past research has found...to be unrelated” -> “past research has not found a relation between...” (no evidence of effect is not evidence for no effect)

**Reviewer #3:** This is an interesting paper that has the potential to make a contribution to the field.

The introduction covers a lot of ground. It would be nice to see this restructured in a way that is focused less on the presentation of summaries of individual constructs, and more in line with an integrated argument that motivates the study. This is likely to be a restructuring rather than a re-writing of the introduction.

I appreciated the pre-registered nature of the project, which is rarely seen in this area of work. Setting the power analysis to 80% is the minimum that would be acceptable in published work and, while sufficient, does limit the rigor of the research. That said, the final sample is above this minimum threshold. Many of the measures used in the study were self-produced, which may limit the independent rigor and/or validity of the inferences drawn. I appreciated the efforts to present some validation of these measures, though this was not always convincing.

I was unclear as to why CFAs were conducted on the scales included in the study. Are there concerns about their factor structure? It felt like these analyses, if needed, could be presented in a supplement. In terms of the SEM, I found the pre-registration document difficult to follow, and the set-up of the latent and observed variables felt a little convoluted. This made it difficult to assess whether the planned analytic approach was followed. The authors report a deviation from the hypothesized model, which led to them removing two variables which, at face value, feel incredibly important (and have been identified as such in prior work). I cannot see any justification for the removal of these variables other than the model fit of the SEM. It would be nice to see some discussion of this deviation from the planned analysis, as the removal of psychological wellbeing and anticipated stigma is inconsistent with prior work in this area.

The conclusions, as presented, make logical sense and I think that there is a role for these data in informing prevention service design. However, the inferences are only as strong as the measures used are valid. Given the self-produced nature of many of the measures used in the study, some caution is required in the interpretations. The authors also acknowledge the questionable psychometrics of some scales (e.g., the Attitudes Toward Seeking Professional Psychological Help Scale) and advise against their future use. In the case of the Attitudes Toward Seeking Professional Psychological Help Scale, the explicit statements about both past research (not finding adequate psychometrics) and future use (advising against this), casts doubt over the validity of what is a (or perhaps even the) fundamental variable in their model. This does limit the extent to which trust can be had in the results of this paper.

Minor points:

- Consider removing statistical details from the abstract. These made it difficult to follow, and could obscure the potential importance of the work

6. PLOS authors have the option to publish the peer review history of their article (what does this mean?). If published, this will include your full peer review and any attached files.

Reviewer #1: No

Reviewer #2: No

Reviewer #3: No

---

## [Author Response · Author response to Decision Letter 0]

15 Jul 2024

I have uploaded the response to each reviewer and editor comments as a word file.

---

## [Decision Letter · Decision Letter 1]

9 Sep 2024

PONE-D-24-03768R1Stigma processes, psychological distress, and attitudes toward seeking treatment among pedohebephilic peoplePLOS ONE

Dear Dr. Jahnke,

Thank you for submitting your manuscript to PLOS ONE. After careful consideration, we feel that it has merit but does not fully meet PLOS ONE’s publication criteria as it currently stands. Therefore, we invite you to submit a revised version of the manuscript that addresses the points raised during the review process.

The reviewers felt that the manuscript was much improved in this revision. However, Reviewer #2 still raises some points that should be corrected to make the manuscript suitable for publication. This is a minor revision, another small effort.

We look forward to receiving your revised manuscript.

Kind regards,

Stefano Federici, Ph.D.

Academic Editor

PLOS ONE

Journal Requirements:

Additional Editor Comments:

The reviewers felt that the manuscript was much improved in this revision. However, Reviewer #2 still raises some points that should be corrected to make the manuscript suitable for publication. This is a minor revision, another small effort.

Reviewers' comments:

Reviewer's Responses to Questions

**Comments to the Author**

1. If the authors have adequately addressed your comments raised in a previous round of review and you feel that this manuscript is now acceptable for publication, you may indicate that here to bypass the “Comments to the Author” section, enter your conflict of interest statement in the “Confidential to Editor” section, and submit your "Accept" recommendation.

Reviewer #1: All comments have been addressed

Reviewer #2: (No Response)

Reviewer #3: (No Response)

2. Is the manuscript technically sound, and do the data support the conclusions?

Reviewer #1: Yes

Reviewer #2: Yes

Reviewer #3: Yes

3. Has the statistical analysis been performed appropriately and rigorously? 

Reviewer #1: Yes

Reviewer #2: Yes

Reviewer #3: Yes

4. Have the authors made all data underlying the findings in their manuscript fully available?

Reviewer #1: Yes

Reviewer #2: Yes

Reviewer #3: Yes

5. Is the manuscript presented in an intelligible fashion and written in standard English?

Reviewer #1: Yes

Reviewer #2: Yes

Reviewer #3: Yes

6. Review Comments to the Author

Reviewer #1: All my comments have been addressed. I could push back against certain interpretations, but I am satisfied with the article as it is.

Reviewer #2: I would like to disclose that between the two review rounds, an event has occurred between one of the authors and me that could be perceived as creating a conflict of interest. I feel that I can review the article objectively nevertheless. I write this before starting to read the revision to make sure it can be factored in in case I’ll make critical remarks.

In my view, the authors did a great job to address my points, and, as far as I can judge, the points raised by the other two reviewers. I see still a need to make changes regarding issues 6 and 20 (which corresponds to a point raised by Reviewer 1) below, and I feel a little at a loss what the authors did regarding issue 16. Regarding all others, I either fully agree with the authors or have just some minor nagging. With these very few changes, I feel the article could make a great contribution to the field.

Reviewer 1 page 92 bottom of the reply letter, related to major issue 20 below: I would like to make an additional suggestion regarding this discussion with Reviewer 1: I see a possible problem by dividing therapists (ourselves as well as in the view of MAPs) into ‘good’ and ‘bad’ therapists, which is probably not better than dividing MAPs in similar categories. In my view, the very wide majority of therapists has best intentions, it’s just that they are human, too, with possibly difficult pasts. So I’d suggest to add that a campaign targeted at pedehebephilic persons should include suggestions how to communicate the situation to the therapist in a way that he or she can cope best and provide support most successfully.

Regarding all issues including the WLSMV estimator, I think the authors did a great job in explaining why they removed it.

Re major issue 1 (original line 217), the new version is great. Very minor, the scale ranging from 1 to 101 would be a 101-point scale, not a 100-point scale; one of the two numbers seems to be in error.

Re major issue 2 (297), I understand the reasoning and agree. You should add the missing treatment explicitly, I.e., I suggest “...ML estimation. The preregistered...” -> “... ML estimation. Using listwise deletion, the preregistered...”.

Re major issue 3 (316), the authors in my view removed the problems successfully.

Re major issue 4 (factor scales strategies): I like the changes. Minors: “...scales function adequately...” -> “...scales worked adequately well...”, “...above .80 and...” -> “... above .80, and...”, “...of CFI = .90., which...” -> “... of CFI = .90, which...” (a dot too many), “...indicated that...” -> “... we used as a threshold to conclude that...”

Re major issue 5 (line 321): I see the point is obsolete now.

Re major issue 6 and 7 (Table 1): I still feel that it seems odd to go to average scores here when you put so much work previously into setting up a good factor model, and that factor models (as I read the weights) fairly strongly says that average scores are not adequate. Sorry if I state the obvious here, but just to be sure we’re on the same page, an average score represents a factor score under the assumptions that all loadings are equal. This is why, given the unequal loadings in your factor model, it seems contradictory in itself suddenly reporting the average score here. I feel your aim of providing the reader with an idea about the central tendency might be better served by reporting the factor scores (lavaan produces those for you, I think; if not, Onyx does). It’s still bivariate analysis even if both variates are latent, so either way, a full SEM shouldn’t violate the preregistration (and preregistration in general never should keep us from thinking about improvements, that would be a negative outcome of preregistration).

Re major issue 8 and 9 (Table 1): The solution the authors chose seems reasonable to me.

Re major issue 10 (368): Also seems a good solution sufficient to me.

Re major issue 11 (369): Okay; not ideal, I’d suggest to go with a Bayesian interpretation here, but at least it’s formally correct now, even though I fear the null result may still be miss-understood as confirming the null by most readers.

Re major issue 12 and 13 (369): This seems a more than thorough solution to the problem.

Re major issue 14 (Table 2): Obsolete now.

Re major issue 15 (430): Seems good enough to me.

Re major issue 16 (437): Okay about the ML estimator of course, but that was only a side remark by me, the main point was about reducing to the openess factor. Sorry if I missed that in your reply in case it was included somewhere above, but how did you change the treatment of the factors?

Re major issue 17 and 18(Table S10): obsolete!

Re major issue 19 (446): Also obsolete mostly, the minor is corrected.

Re major issue 20 (498): See remark regarding Reviewer 1’s concern about the same point; I like what you did, but I’d suggest to add the component of “helping the therapist along”.

As I thinking this over again, as much as I hate to say this: I do suggest that you add explicitly “In very rare case, visiting a psychotherapist can result in a life-threatening situation.” It’s a rare exception and it shouldn’t keep MAPs from seeing psychotherapists when all is said and done, but so is CSA, and it shouldn’t keep children from interacting with MAPs, but that doesn’t mean we should not be honest about the existence of the risk. You may of course add a big “but” afterwards!

Re minor (146 in the original manuscript): Your assumption was considered general knowledge until very recently. However, newest literature suggests that contrary to what we believed before, it may in fact be possible to deduce causality from cross-sectional observational data when not all variables involved are normally distributed. But this is cutting edge, and your data is normal, so unless you’re really strict about not saying anything that is very technically wrong, just leaving this as it is is just fine in my view.

Re author replies to Reviewer 3, I have no comments to add.

Reviewer #3: The manuscript is much improved and the authors have done a good job in addressing most reviewer comments.

7. PLOS authors have the option to publish the peer review history of their article (what does this mean?). If published, this will include your full peer review and any attached files.

Reviewer #1: No

Reviewer #2: No

Reviewer #3: No

---

## [Author Response · Author response to Decision Letter 1]

1 Oct 2024

Our responses are included in the "letter to the reviewers", which we have uploaded previously.

---

## [Decision Letter · Decision Letter 2]

7 Oct 2024

Stigma processes, psychological distress, and attitudes toward seeking treatment among pedohebephilic people

PONE-D-24-03768R2

Dear Dr. Jahnke,

We’re pleased to inform you that your manuscript has been judged scientifically suitable for publication and will be formally accepted for publication once it meets all outstanding technical requirements.

Kind regards,

Stefano Federici, Ph.D.

Academic Editor

PLOS ONE

Additional Editor Comments (optional):

Reviewers' comments:

Reviewer's Responses to Questions

**Comments to the Author**

1. If the authors have adequately addressed your comments raised in a previous round of review and you feel that this manuscript is now acceptable for publication, you may indicate that here to bypass the “Comments to the Author” section, enter your conflict of interest statement in the “Confidential to Editor” section, and submit your "Accept" recommendation.

Reviewer #2: (No Response)

2. Is the manuscript technically sound, and do the data support the conclusions?

Reviewer #2: Yes

3. Has the statistical analysis been performed appropriately and rigorously? 

Reviewer #2: Yes

4. Have the authors made all data underlying the findings in their manuscript fully available?

Reviewer #2: Yes

5. Is the manuscript presented in an intelligible fashion and written in standard English?

Reviewer #2: Yes

6. Review Comments to the Author

Reviewer #2: Thanks for the revision, I think the article is really great work and will help readers in the field tremendously. I still disagree with issue 20: I am strongly convinced that in close-to-all cases, therapist will not break confidentiality to outsiders when MAPs disclose their sexuality in therapy. At the same time, I know for a fact that it does happen. I also know for a fact that in close-to-all cases, interactions between MAPs and children are not abusive. At the same time, I know for a fact that it does happen. In my view, we gain more intellectually and for the best of patients if we are honest about all four facts, even if we then add that negative outcomes are less probable than, say, getting wounded by a hailstone when walking in the rain. However, that is an opinion about communication at the end of the day, so since you seem set on not saying "therapists can (very rarely) pose a significant danger for the patients" explicitly, that in my view can't be a reason to not publish the article.

7. PLOS authors have the option to publish the peer review history of their article (what does this mean?). If published, this will include your full peer review and any attached files.

Reviewer #2: No

---

## [Editor Report · Acceptance letter]

14 Oct 2024

PONE-D-24-03768R2 

PLOS ONE

Dear Dr. Jahnke, 

I'm pleased to inform you that your manuscript has been deemed suitable for publication in PLOS ONE. Congratulations! Your manuscript is now being handed over to our production team.

Kind regards, 

on behalf of

Prof. Stefano Federici 

Academic Editor

PLOS ONE